# Analysis of the Pulpal Blood Flow Microdynamics during Prosthetic Tooth Preparation Using Diamond Burs with Different Degrees of Wear

**DOI:** 10.3390/dj12060178

**Published:** 2024-06-07

**Authors:** Edmond Ciora, Mariana Miron, Diana Lungeanu, Andreea Igna, Anca Jivanescu

**Affiliations:** 1Department of Oral Rehabilitation and Dental Emergencies, Faculty of Dentistry, “Victor Babes” University of Medicine and Pharmacy, P-ta Eftimie Murgu 2, 300041 Timisoara, Romania; ciora.edmond@umft.ro; 2Digital and Advanced Technique for Endodontic, Restorative and Prosthetic Treatment TADERP, 300070 Timisoara, Romania; jivanescu.anca@umft.ro; 3Interdisciplinary Research Center for Dental Medical Research, Lasers and Innovative Technologies, 300070 Timisoara, Romania; 4Center for Modeling Biological Systems and Data Analysis, “Victor Babes” University of Medicine and Pharmacy, 300041 Timisoara, Romania; dlungeanu@umft.ro; 5Department of Functional Sciences, “Victor Babes” University of Medicine and Pharmacy, 300041 Timisoara, Romania; 6Department of Pediatric Dentistry, Pediatric Dentistry Research Center, Faculty of Dental Medicine, “Victor Babes” University of Medicine and Pharmacy, 300041 Timisoara, Romania; igna.andreea@umft.ro; 7Department of Prosthodontics, Faculty of Dentistry, “Victor Babes” University of Medicine and Pharmacy, Eftimie Murgu Square No. 2, 300041 Timisoara, Romania

**Keywords:** prosthetic tooth preparation, diamond burs, degrees of wear, pulp blood flow, Laser Doppler flowmetry

## Abstract

Pulpal modifications taking place during prosthetic tooth preparation using worn-out burs may represent a risk for the vitality of the dental pulp. The aim of this in vivo study was to evaluate whether the wear of diamond burs has an influence on the vascular microdynamics at the level of the dental pulp, during vertical preparation for zirconia crowns. The study was performed with a split-mouth design and included 32 vital permanent monoradicular teeth (20 maxillary and 12 mandibular), from six subjects, aged between 20 and 50 years. The teeth were randomly assigned to two study groups of 16 teeth each. For prosthetic preparation, new burs were used in the first group, and burs at their 5th use were used in the second group. Four consecutive determinations of the pulpal blood flow by Laser Doppler flowmetry (LDF—laser Doppler MoorLab instrument VMS-LDF2, Moor Instruments Ltd., Axminster, UK) were taken for each tooth included in the study: before the preparation (control values), immediately, at 24 h, and at 7 days after the prosthetic preparation. A four-way ANOVA statistical analysis was applied to analyze the effect of four considered factors (bur wear degree, time of measurement, tooth number, and tooth location) on the pulpal blood flow (PBF). A significant increase in pulpal blood flow compared to the baseline was recorded immediately after preparation (*p* < 0.01), at 24 h (*p* < 0.01), and at 7 days (*p* < 0.05) in both groups, but more pronounced in the case of burs at the 5th use. The blood flow was significantly higher in upper jaw teeth, irrespective of the measurement time. In conclusion, the use of worn-out diamond burs produces lasting modifications in the pulpal blood flow of teeth that undergo prosthetic crown preparation. ISRCTN registry: ISRCTN49594720.

## 1. Introduction

Evaluation of dental pulp vitality related to prosthetic preparation represents an important objective in the dental practice in order to improve the restoration’s prognosis [1,2]. Therefore, in clinical practice, assessment of the dental pulp status is a key step in the diagnosis of pulpal disease; the accuracy of pulp diagnosis relies on corroborating data obtained from clinical examination, radiographic examination, pulp testing, and anamnestic data [3]. The conventionally used pulp testing methods are based on sensibility testing, being highly subjective because of their direct dependance on the patient’s perception of pain [4,5]. Research from the past decades has highlighted the importance of pulpal microcirculation for maintaining vitality of the teeth [6,7,8,9,10]; therefore, pulp tests that address blood perfusion—namely the vitality tests—have gained the attention of researchers. Non-invasive and independent of the patient’s response during the assessment, vitality tests such as Laser Doppler flowmetry (LDF) and pulse-oximetry have proven their superiority over sensibility tests in multiple clinical scenarios—in recently traumatized teeth [11,12,13], following therapeutic interventions like vital pulp therapy and revascularization [10,14], in orthodontic patients [15], etc. The LDF technique, first described in the scientific literature in 1986 by Gazelius et al. [6], is still the only available method that enables continuous, real-time observation of pulpal flux and detection of its dynamic changes. Also, it is the only option that allows quantitative pulpal blood flux assessment [16] and offers data that assess the progression of vascular flow dynamics before the appearance of clinical signs [17].

Prosthetic preparation is an invasive process which involves a substantial amount of hard dental structure being removed using high-speed rotary instruments, with possible adverse effects on the adjacent pulp or periodontal tissues [18]. The instruments typically employed for this procedure are the diamond burs, as they exhibit superior cutting efficiency and durability compared to other types of burs [19]. Studies have shown that their cutting efficiency diminishes with each subsequent use [20] and may be attributed to wear and the loss of diamond particles and binder material [21], the decline being most pronounced following the initial use. Consequently, worn-out burs may necessitate increased pressure during tooth preparation, potentially leading to unwanted heat generation and therefore posing a higher risk for the dental pulp [19,20]. Similarly, the use of coarse diamond burs results in more pronounced temperature increase within the pulpal chamber during tooth preparation [22]. The consequences of increasing the intrapulpal temperature were evaluated by Zach and Cohen, and they showed that an intrapulpal increase of 5.5 °C above normal body temperature is likely to cause irreversible alterations of the pulp tissue, such as inflammation, abscess, and necrosis. Reversible alterations were also noted during intrapulpal temperature rise, like thinning of predentin, edematous stroma, and aspiration of odontoblast nuclei into the dentinal tubules [23]. In this context, according to Lau et al., a 3 °C temperature rise is considered nowadays the threshold to avoid producing these effects [24]. Temperature increase in the dental pulp prolonged for more than 1 min triggers structurally irreversible pulp tissue damage [25]. 

The changes in the pulpal blood flow related to crown preparation were first investigated in 1992 by Kim et al., who emphasized the crucial role of water irrigation during the procedure, for the health of the underlying pulp tissue [26]. Following tooth preparation for prosthetic restorations, the remaining dentin thickness is strictly correlated to the severity of degenerative parameters, while different preparation techniques do not demonstrate clearly different endodontic reactions [27,28]. A crucial aspect of successful restorative procedures is to minimize additional irritation to the pulp and avoid disruption to normal pulpal healing. Numerous researchers have asserted that vital teeth exhibit greater resistance to bacterial infiltration into dentinal tubules compared to non-vital teeth [2,29,30] and more resilience to fracture than endodontically treated teeth [30]. This suggests a significant role played by the vital pulp in this process.

Despite the importance of this topic, there are limited studies available which investigate the influence of prosthetic tooth preparation on pulpal blood flow. While various experimental setups of simulating intraoral environment have been employed in most previous studies [19,24,31,32,33,34,35], variables such as pulp blood flow, intraoral temperature, and intraoral humidity are still difficult to manage under in vitro conditions [24]. Furthermore, the research on this topic carried out in vivo is limited. That is why, in order to achieve results that closely resemble clinical reality, studies should be designed to incorporate as many variables as possible that are encountered in current practice.

The aim of this in vivo study was to evaluate whether the wear of diamond burs has an influence on the vascular microdynamics at the level of the dental pulp, during vertical preparation for zirconia crowns. Also, the evolution of the pulp blood flow in the four determination moments was analyzed.

The null hypothesis states that the diamond burs with a certain degree of wear used for zirconia crown preparation do not produce statistically significant different changes in the blood flow at the dental pulp level compared to new diamond burs.

## 2. Materials and Methods

### 2.1. Sample Selection, and Ethical Aspects

The study was performed at the Oral Rehabilitation and Dental Emergencies Clinic, Faculty of Dental Medicine, Victor Babes University of Medicine and Pharmacy, Timisoara. The research design and protocol received approval from the Research Ethics Committee (approval number 40 of 4 April 2022) of the Victor Babes University of Medicine and Pharmacy in Timisoara. Each patient signed an informed consent after the procedures had been completely explained to them. The study included thirty-two (*n* = 32) vital permanent monoradicular frontal teeth (incisors and canines), twenty from the maxilla (8 central incisors, 8 lateral incisors, and 4 canines) and twelve from the mandible (4 central incisors, 4 lateral incisors, and 4 canines), from 6 subjects, aged between 20 and 50 years (mean age 38, 4 males, 2 females), selected from the Oral Rehabilitation and Dental Emergencies Clinic patients. The inclusion criteria specified: healthy patients, without general or local conditions, or treatments that could influence vascular hemodynamics; non-smoker patients; patients presenting at least four maxillary or mandibular monoradicular frontal teeth, symmetrically arranged with respect to the median line of the arch; vital teeth, with no clinical and radiographical signs of pulpal inflammation; teeth free of carious lesions or prosthetic restorations. 

The exclusion criteria included: patients with any general or local condition, or medication that could influence vascular dynamics; smokers; pluriradicular teeth; teeth with carious lesions; teeth with fillings of class IV, V; teeth with prosthetic restorations; teeth with any form of reversible or irreversible pulp inflammation; teeth treated with bleaching agents within the past six months; patients with recent orthodontic treatment; patients with locomotory or psychoemotional disorders;

Radiographic examination and electrical pulp stimulation confirmed that all teeth included in the present study were vital and presented with no signs of pulpal inflammation (normal response to the electrical pulp stimulation, no radiographic evidence of periapical lesions, and negative response to the percussion test).

### 2.2. Study Design

A randomized, single-blinded clinical study was performed with a split-mouth design by using the mid-sagittal plane between the central incisor teeth. The study was designed as a mixed factorial setup, combining the between-group treatment factor with the within-subject variable of repeated measures in time.

### 2.3. Materials and Study Protocol

The selected teeth were prepared for full coverage monolithic zirconia prosthetic restorations.

As such, the teeth were randomly assigned to two study groups of 16 teeth each:Gr. A (*n* = 16): selected to be prepared with new burs (first use);Gr. B (*n* = 16): selected to be prepared with burs at their fifth use;

We chose the minimum threshold of five uses of the bur as there is evidence in the scientific literature that recommends changing the diamond burs after 5 teeth preparations at most [19,36]. Table 1 shows the group distribution of the sample elements, according to the study design.

Four consecutive determinations of the PBF were taken for each tooth included in the study: before the preparation (control values), immediately, at 24 h, and at 7 d after the prosthetic preparation for crown (Figure 1).

The diamond burs included in the study were used for the first time (new) in the 1st group and for the 5th time in the 2nd group. That is, the burs had already been previously used four times for other dental preparations, at five minutes per use.

For each tooth, a three-tier diamond depth cutter (Meisinger, depth cutter 834, FG 806 314 552 524, L—0.5 mm, Neuss, Germany) with a 0.5 mm thickness was used to create guide grooves on the vestibular surface. Subsequently, the tooth surface was prepared using diamond burs with different levels of wear, as specified by the study design. The same operator performed each preparation, using a consistent cutting technique for five minutes per tooth; the layer of removed hard tissue ranged between approximately 1 and 1.5 mm for the vertical walls, respecting the anatomical characteristics of the dental crowns, and between 1.5 and 2 mm for the incisal edges. The burs used for teeth preparation were from Komet dental, cylindroconical bur 6859 dimensions/sizes: 018; diameter: 1/10 mm; length: 10.0 mm; maximum speed: 300,000; angle: 3.6°, grit: coarse, particle grit size: 151 microns. Regarding the use of burs protocol, each bur was utilized following a standardized procedure. Initially, we used calibrated depth cutters to create orientation grooves, ensuring consistent and precise reduction. Each bur was used to prepare a tooth, sterilized, and then reused. After five preparation cycles and five sterilizations, the burs were considered to be at the next stage of wear, and a new set was used. This process ensured consistent performance and accuracy throughout the study.

Teeth preparation was performed using the SMARTtorque LUX S619L turbine manufactured by KAVO^®^ (Kavo Dental GmbH, Bismarckring 39, Biberach, Germany), which has a built-in push-button mechanism for securing the drill bit and a 4-hole spray design to ensure efficient cooling during operation. This handpiece was linked to the dental unit via a MULTIflex™ LUX connection by KAVO^®^. It had a maximum bur rotation speed of 400,000 rpm. The preparations were performed under a water flow rate of 50 mL/min, and the cooling water reservoir was filled with water at a temperature of 20 °C ± 0.5. Additionally, we used a standard High-Volume Evacuation (HVE: Durr Dental, VMS 600, Suction power of 300 L/min per unit) suction device to efficiently manage fluid control during tooth preparation. This device was operated by a trained dental assistant to ensure consistent fluid removal and to maintain a clear operative field throughout the procedure. To maintain hygiene standards, the turbine underwent sterilization at a temperature of 135 °C.

### 2.4. LDF Device

In order to assess LDF signals from dental pulp, we used a MoorLab laser Doppler device for general medical use (laser Doppler MoorLab instrument VMS-LDF2, Moor Instruments Ltd., Axminster, UK) and a straight optic probe VP3 with a length of 10 mm, built to be used on the oral mucosa/teeth. The MoorLab laser Doppler monitor (Moor Instruments) uses laser radiation generated by a semi-conductor laser diode operating at a wavelength of 780 ± 10 nm and a maximum accessible power of 1.6 mW. The programmed bandwidth of the recorded laser Doppler signal was 20 Hz–20 kHz, while sampling frequency displayed a value of 40 Hz. Probe calibration was performed according to the instructions of the manufacturer. A personal computer system for collecting and processing the data was also needed.

### 2.5. The Study Protocol

On the first appointment, after signing the informed consent, the teeth included in the study were selected. Furthermore, the LDF probe holder was created as follows: around the gingival area of every tooth involved in the study, a light-cured periodontal liquid dam Ultradent^®^ LC Block-Out Resin (Ultradent Products, South Jordan, UT, USA) was applied on a radius of 3–4 mm. Subsequently, a double silicone impression using Kit Optosil Comfort Putty and Xantopren Comfort Light, Haereus (Heraeus Kulzer, GmbH Leipziger Straße 2, Hanau, Germany) was realized, so that the impression covered two teeth on each side of the tested tooth. This probe holder served for the acquisition of the laser Doppler signals of the tested teeth in all of the four moments of measurement (Figure 2). After decontaminating the impression, a drill of 1.5 mm in diameter was used to create a hole from the vestibular (buccal) to the oral side of the impression, perpendicularly to the cervical third of the tooth involved in the study, at a 3 mm distance from the marginal gingiva. In order to ensure the reproducibility of the laser Doppler signal acquisition, a guiding mark that permitted its placement in the same position for each testing was set on the fiber. Then, the tooth’s surface was cleaned, and the impression was repositioned. The flux signals for each patient were acquired for one and a half minutes during each measurement. The laser Doppler signal acquisition technique was performed according to our previous studies [9,17].

On the second appointment, the teeth were prosthetically prepared in order to receive monolithic zirconia crowns, and the blood flow signals were also recorded immediately after preparation. The LDF signal was also assessed at 24 h and at 7 days after prosthetic preparation. We mention that before each recording, the liquid dam was applied around the evaluated teeth, in order to isolate the pulpal Doppler signal and reduce contamination with signals from the marginal periodontium (Figure 3).

Between the appointments, teeth were protected by provisional acrylic crowns, in order to eliminate other factors that may influence the results of the testing, such as temperature, direct occlusal forces applied on the polished teeth, and contamination with bacteria from the oral cavity. The provisional restorations were acrylic crowns made using Protemp™ 4 Temporization Material (3 M Deutschland GmbH, Neuss, Germany). These crowns were cemented with Temp-Bond NE (Kerr Italia S.R.L., Via Passanti, 174 Scafati (SA), Italy). The provisional restorations were applied immediately after the tooth preparation process, to avoid further stimulation of the dental pulp by environmental factors, thus limiting their influence on the following PBF measurements. The method of fabrication of the provisional restoration was direct (chairside), using a matrix made prior to prosthetic preparation.

This study was registered in the ISRCTN registry (Unique ID: ISRCTN49594720) and is available on https://www.isrctn.com (accessed on 4 June 2024) (doi:10.1186/ISRCTN49594720).

### 2.6. Statistical Analysis

Four-way ANOVA was used for analysis: pulp blood flow values (numerical variable) and four factors (categorical variables). Post hoc comparisons were conducted according to the Tukey procedure. The factors were: degree of bur wear, time of measurement, tooth number, tooth location (upper or lower jaw).

The statistical analysis was conducted at a 5% level of statistical significance, and all reported probability values were two-tailed. The analysis was performed using the statistical software IBM SPSS v. 20.0. (Armonk, New York, NY, USA).

## 3. Results

The descriptive statistics included the mean and standard deviation of pulpal blood flow for each combination of categorical variables (namely, the four considered factors) and are presented in Table 2.

In Figure 4, the evolution of pulpal blood flow in the four moments of time is represented comparatively, for upper maxillary and mandibular teeth prepared with burs at their first (left) and fifth use (right), respectively.

The results for four-way ANOVA analysis are shown in Table 3.

Post hoc multiple comparisons for the measurement time with Tukey procedure are presented in Table 4.

## 4. Discussion

In the conditions of the present study, the results of the four-way ANOVA analysis (Table 3), applied to analyze the effect of the four considered factors (the degree of wear of the burs, time of measurement, tooth number, and tooth location) on the pulpal blood flow, showed that the measurement time (baseline, immediate, at 24 h, at 7 days after prosthetic preparation) and tooth location (maxilla or mandible) factors had a highly statistically significant effect (*p*-value < 0.001) at *p* < 0.01, while the bur wear factor had a statistically significant effect at *p* < 0.05. The tooth number factor was not statistically significant in the ANOVA analysis.

Furthermore, the effect of the time factor on the PBF values was analyzed through post hoc multiple comparisons, conducted according to the Tukey procedure. The results, presented in Table 4, show highly statistically significant changes in PBF values immediately after preparation, at 24 h, and at 7 days, compared to the initial moment (*p*-value < 0.001, *p*-value < 0.001, and *p*-value = 0.008) at a significance threshold of *p* < 0.01. By analyzing Figure 1, it is observed that there is a significant difference in signal levels acquired before preparation (m = 7.8 PU) and at 24 h after (m = 15.2 PU) and also the loss of the pulsatile nature of the laser Doppler signal. This aspect is attributed to the irritative effect following the tooth preparation. Even if patients did not report any painful symptoms clinically at 24 h, the laser Doppler signal still revealed changes in vascular microdynamics. Additionally, highly statistically significant differences were found between the values of pulpal blood flow recorded at 24 h vs. 7 days after preparation (*p*-value < 0.001) at *p* < 0.05 (Table 4). This evolution of pulpal blood flow at the four recording moments, considered in the study design, can be easily observed in the graphics in Figure 4.

Our study results regarding the significant changes in PBF after prosthetic tooth preparation, recorded at different time points (immediately, at 24 h, after 7 days), are in line with the study led by Uğur-Aydın et al. in 2024 [37]. In contrast to our results, a 2016 study led by Sukapattee M. et al. found no change in PBF following full-crown preparation, neither immediately after the procedure nor 1 and 7 days later [38]. 

The analysis of the effect of the degree of wear of the burs used for preparation on the evolution of PBF showed a statistically significant effect with a *p*-value = 0.013 at a significance threshold of *p* < 0.05 (Table 3). 

Our results indicated that the location of the tooth (maxilla or mandible) had a highly statistically significant effect on pulpal blood flow (PBF) (*p* < 0.001). Specifically, the mandibular teeth exhibited a more pronounced increase in PBF immediately after preparation compared to the maxillary teeth. This difference may be attributed to the variations in tissue density and vascular supply between the maxilla and mandible. These are consistent with those of Gumus et al., 2014 [39] and suggest that mandibular teeth might be more sensitive to the mechanical stress of tooth preparation, possibly due to their anatomical and physiological characteristics. Therefore, it is essential to consider these differences in clinical practice to ensure optimal outcome in prosthetic treatments.

Comparing the flow evolution between the baseline moment and immediately after preparation, both in the maxilla and mandible, a more pronounced increase is observed in the case of burs at the 5th use (Figure 4). This graphical representation, in accordance with the data presented in Table 3, supports the result of the four-way ANOVA analysis. Consequently, the results suggest that preparation with burs at the 5th use (with a certain degree of wear) leads to a more significant increase in PBF than that obtained following the use of new burs, indicating that the degree of bur wear had a greater impact on PBF changes in mandibular teeth.

Therefore, the results obtained under the current study conditions reject the null hypothesis which stated that the different degrees of wear of conventional diamond burs used for zirconia crown preparation do not produce statistically significant different changes in the blood flow at the dental pulp level. 

When interactions between the various factors were explored, the different types of teeth (comprising both tooth number and location) were proved to experience significantly different effects of the bur during prosthetic tooth preparation. Additionally, this association with the degree of burs’ wear produced highly significant differences in PBF.

Both when using new burs and when using those for the fifth time (with wear), a significant increase in pulpal blood flow compared to the baseline was recorded immediately after preparation (*p* < 0.01), at 24 h (*p* < 0.01), and at 7 days (*p* < 0.05).

The PBF values recorded at 7 days exhibit a significantly high decrease compared to the values recorded at 24 h (*p* < 0.01), yet they remain significantly higher than the baseline values (*p*-value = 0.008).

The microdynamics of PBF exhibited a similar upward trend immediately after preparation, accentuated immediately post-grinding, with a downward trend after 7 days, maintaining a level higher than the baseline. PBF values vary significantly depending on the recording moment and are significantly influenced by the degree of wear of the burs.

Regarding hard tissue reduction, in order to safely accommodate zirconia crowns, according to many studies [40,41,42,43,44,45] the preparation requires a minimum of 0.3, ideally 1.0 to 1.5 mm space to make room for the crown walls’ thickness, along with an incisal reduction of 1.5 to 2 mm. Jasim M. et al. [41] and Loi et al. [40] found that the shoulderless margin design yields more favorable results compared to a slight chamfren design across all thicknesses. The same preparation had been carried out in our study, and it included allocating approximately 1–1.5 mm of space to accommodate the thicker walls, respecting the anatomical characteristics of the dental crowns and with an occlusal reduction ranging from 1.5 mm up to 2 mm, guided by occlusal parameters.

Concerning the issue after how many uses a bur should be changed, several studies found that the cutting efficiency of diamond burs diminishes with each subsequent use, most pronouncedly following the initial use, regardless of the type of instrument [20,21,46]. The cutting efficiency is influenced by several factors, including grit size [20,46,47], coolant flow [33,48], load applied by the operator, bur shape, tooth structure [20,26], repeated cleaning and sterilization process [20,49,50], use of the electric micromotor vs. air turbine handpiece and cutting technique [24,48], the number and location of the coolant ports of the dental handpieces [24,48]. Consequently, there is no consensus on a fixed number of uses/bur. Pilcher et al. [21] recommend, prior to preparation, inspection of the multiple-used burs using magnification. According to the results of our studies, within its limits, we have observed, by using LDF, that even after the 5th use, there were differences on the pulpal blood flow evolution, but without clinical symptoms.

Thus, regularly changing diamond burs is crucial for maintaining their effectiveness [19,36], while prioritizing water cooling is essential for preserving pulp integrity [31]. Our findings are supported by the literature evidence emphasizing the significance of both aspects.

The influence of bur wear was addressed by us in a previous in vitro study [51]. Consequently, the current study represents a natural continuation in the process of discovering the effects on dental pulp due to prosthetic tooth preparation.

### Study Limitations and Future Perspectives

The limitations of our study include the need to improve the study design by expanding the sample size, including pluri-radicular teeth, therefore considering anatomical variations. Additionally, in line with current prosthodontic practice, burs with even more varied degrees of wear must be evaluated, and the quantification of the surface roughness can be performed under laser surface scanning. Future research could explore the reduction in tooth hard tissue by utilizing intraoral scanning and measuring the extent of reduction through the overlaying of scans. Moreover, we acknowledge the fact that no magnification assessment was used to examine the surface load of the multiple used burs as a limitation of this study. 

For future studies, we plan to incorporate a more rigorous evaluation of bur wear, possibly including microscopic analysis and quantifying the impact of bur condition on data variability. However, controlling variables proved to be more challenging in the in vivo setting.

## 5. Conclusions

To sum up, within the limitations of this study, the results showed:−A significant increase in pulpal blood flow, immediately and especially at 24 h post-preparation, both when using new burs and those at their 5th use;−The PBF values recorded at 7 days exhibit a significantly high decrease compared to the values recorded at 24 h, yet they remain significantly higher than the baseline values;−Preparing teeth with burs with a certain degree of wear (on the 5th use) leads to a more significant increase in pulpal blood flow than that recorded after using new burs;−The mandibular teeth exhibited a more pronounced increase in PBF immediately after preparation compared to the maxillary teeth.

## Figures and Tables

**Figure 1 dentistry-12-00178-f001:**
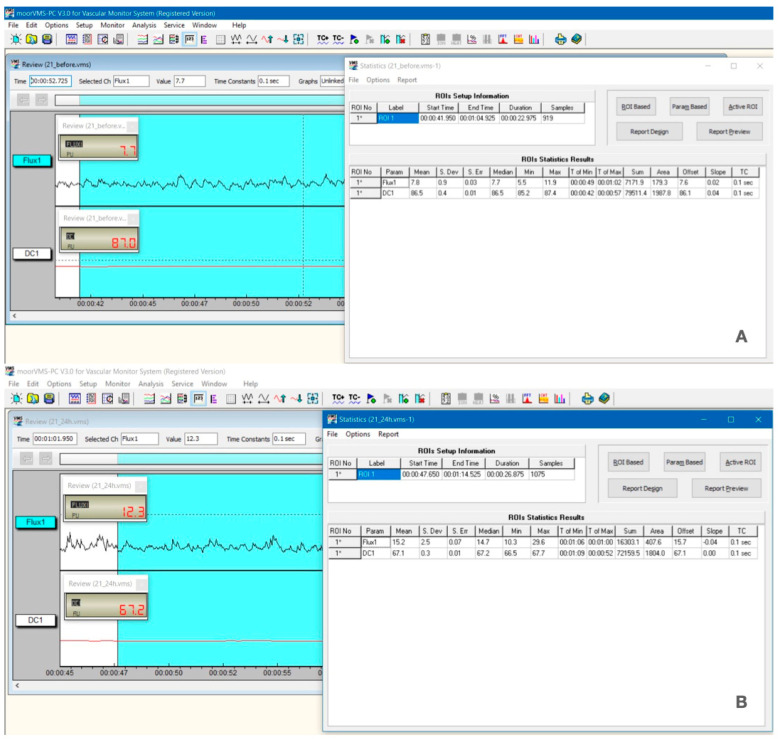
Laser Doppler signal level for tooth 2.1 (patient no. 2), recorded before preparation (**A**) and at 24 h (**B**).

**Figure 2 dentistry-12-00178-f002:**
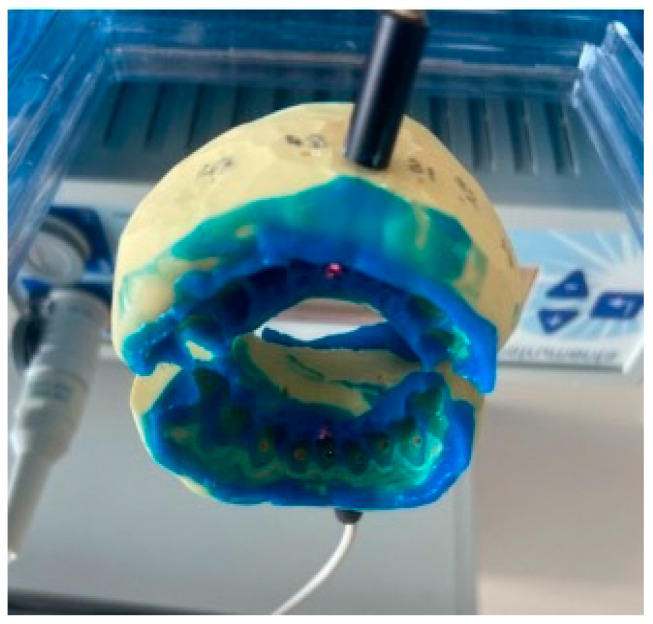
Laser Doppler probe holder.

**Figure 3 dentistry-12-00178-f003:**
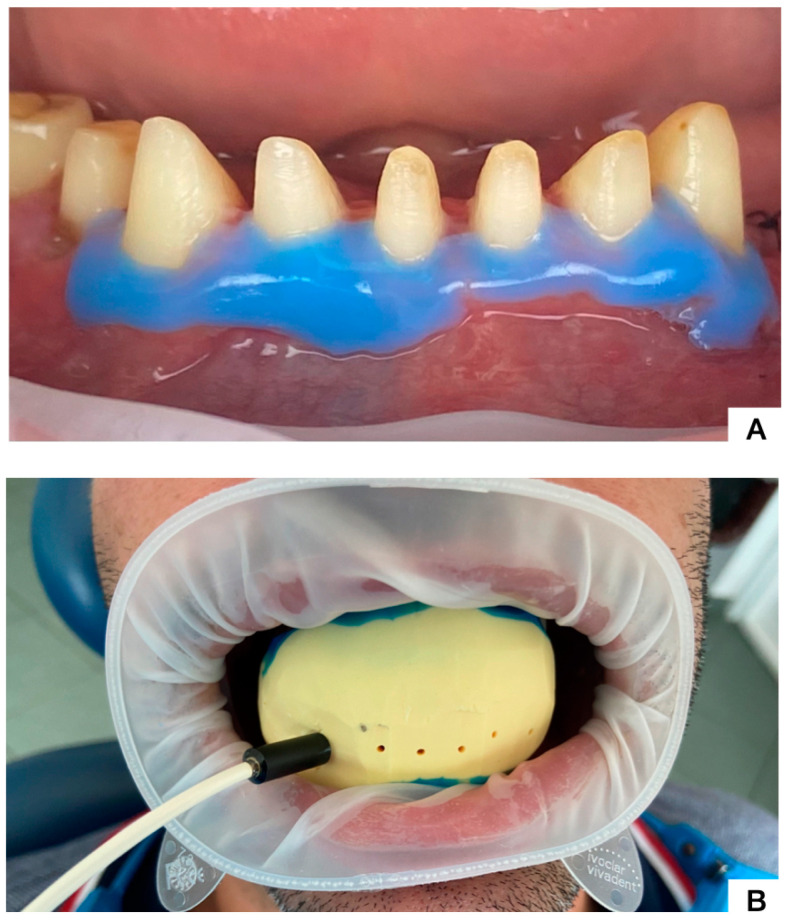
(**A**). Mandibular teeth following prosthetic preparation and isolation of the marginal gingiva using liquid dam; (**B**). LDF probe holder for the pulpal blood flow acquisition from tooth 4.3.

**Figure 4 dentistry-12-00178-f004:**
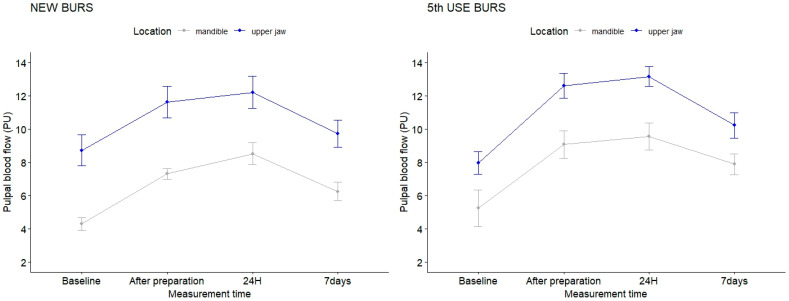
Pulp blood flow in time, for the different combinations of significant factors.

**Table 1 dentistry-12-00178-t001:** The group distribution of the sample elements, according to the study design.

Group	Diamond Burs’ Wear	Sample Size	Teeth	
A	New	*N* = 16	upper jaw	*N* = 10
			mandible	*N* = 6
B	Fifth use	*N* = 16	upper jaw	*N* = 10
			mandible	*N* = 6

**Table 2 dentistry-12-00178-t002:** Descriptive statistics (mean ± standard deviation) for the considered factors.

Burs Wear	Tooth Location	Tooth Number	Baseline	After Preparation	24 h	7 Days
New						
	upper jaw	13 (*N* = 2)	5.45 ± 1.05	8.65 ± 1.3	8.25 ±1.2	6.7 ± 0.9
		12 (*N* = 4)	8.95 ± 1.42	9.62 ± 1.75	12.2 ± 1.97	9.7 ± 1.42
		11 (*N* = 4)	10.12 ± 1.82	12.87 ± 1.95	14.2 ± 2.15	11.27 ± 1.85
		Total *N* = 10	8.17 ± 1.43	10.38 ± 1.66	11.55 ± 1.77	9.22 ± 1.39
	mandible	43 (*N* = 2)	5.3 ± 0.8	7.35 ± 1.05	9.45 ± 2.1	7.05 ± 1
		42 (*N* = 2)	4.05 ± 0.75	7.9 ± 1	8.85 ± 1.05	6.8 ± 1.35
		41 (*N* = 2)	3.5 ± 0.7	6.65 ± 1.05	7.25 ± 1.2	4.85 ± 0.85
		Total *N* = 6	4.28 ± 0.75	7.3 ± 1.03	8.51 ± 1.45	6.23 ± 1.06
5th use						
	upper jaw	21 (*N* = 4)	7.67 ± 1.45	12.35 ± 2.15	12.92 ± 2.02	9.2 ± 1.4
		22 (*N* = 4)	8.87 ± 1.47	13.65 ± 2.5	13.75 ± 2.12	11.25 ± 1.6
		23 (*N* = 2)	6.7 ± 1.2	11 ± 1.9	12.45 ± 1.9	10.2 ± 1.7
		Total *N* = 10	7.74 ± 1.37	12.33 ± 2.18	13.04 ± 2.01	10.21 ± 1.56
	mandible	31 (*N* = 2)	5.3 ± 0.9	10 ± 1.85	10.45 ± 1.5	7.8 ± 1.35
		32 (*N* = 2)	5 ± 0.95	8.05 ± 1.6	8.1 ± 1	7.75 ± 1.1
		33 (*N* = 2)	5.4 ± 0.9	9.15 ± 1.25	10.1 ± 1.15	8.1 ± 1.1
		Total *N* = 6	5.23 ± 0.91	9.06 ± 1.56	9.55 ± 1.21	7.88 ± 1.18

**Table 3 dentistry-12-00178-t003:** Four-way ANOVA analysis results for the considered factors.

	Factor	F-Statistic (df)	*p*-Value
Main effects	Time of measurement	20.146 (3)	<0.001 **
	Burs’ wear	6.492 (1)	0.013 *
	Tooth location	55.541 (1)	<0.001 **
	Tooth number	2.038 (2)	0.137
Two-way interactions	Time * burs	0.427 (3)	0.734
	Time * location	0.188 (3)	0.904
	Time * tooth	0.164 (6)	0.985
	Burs * location	0.393 (1)	0.532
	Burs * tooth	0.774 (2)	0.465
	Location * tooth	6.268 (2)	0.003 **
Three-way interaction	Time * burs * location	0.197 (3)	0.898
	Time * burs * tooth	0.108 (6)	0.995
	Time * location * tooth	0.087 (6)	0.997
	Burs * location * tooth	5.479 (2)	0.006 **
Four-way interaction	Time * burs * location * tooth	0.140 (6)	0.990
Model		3.902 (47)	<0.001 **

Abbreviation: df, degrees of freedom; *, *p* < 0.05 (statistical significance); **, *p* < 0.01 (high statistical significance).

**Table 4 dentistry-12-00178-t004:** Post hoc multiple comparisons for the measurement time.

Comparisons (Tukey Procedure)		*p*-Value
Time of measurement	Baseline vs. After preparation	<0.001 **
	Baseline vs. 24 h	<0.001 **
	Baseline vs. 7 days	0.008 **
	After preparation vs. 24 h	0.636
	After preparation vs. 7 days	0.015 *
	24 h vs. 7 days	<0.001 **

*, *p* < 0.05 (statistical significance); **, *p* < 0.01 (high statistical significance).

## Data Availability

The data can be accessed by the researchers who participated in this study and are not publicly stored on servers.

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
