# Peer review of "Analysis of the Pulpal Blood Flow Microdynamics during Prosthetic Tooth Preparation Using Diamond Burs with Different Degrees of Wear"

_dentistry, 2024, doi:10.3390/dj12060178_

Round 1
Reviewer 1 Report
Comments and Suggestions for Authors
Row 33 should provide the abbreviation PBF
ROW 116-117 tangential preparation; it should be used verical preparation term. Stating for zirconia crown do not provide information about the quantity of reduction, this is influenced by the type of crown, tooth and type of zirconia. Authors should state the reduction quantity.
row 129-130: which teeth? The number of the tooth implies different volume and the quantity of reduction for a given crown and material is the same. If teeth are very different in size the preparation would induce different pulp reaction.
row 132-133: the teeth are intact? You stated no carious lesion? Were there filings?
row 134-138 in the introduction section you stated that electrical stimulation are highly subjective (row 52) yet you use it as an inclusion criteria.
Row 145: state the reduction parameters and how we assessed how much dental tissue was taken off
row 147-149 type of teeth
row 163-165 you should explain better how the burs were used: not only the time
row 220: what type of acrylic crowns? How were they made? What material was used for cementation? When were they applied? all those factors may influnce the following measurements
in discussion section: more info linked to the difference in between the maxilly and mandibular teeth.
which is the novelty of the study?
when it should recommended to change the burs?
ref 23 is incomplete
Author Response
Dear Reviewer 1,
We sincerely thank you for rigorously reviewing of our material and for your detailed feedback, which has helped us substantially to improve our manuscript. In what follows we have taken into consideration your guidelines as it can be seen in the attached answer.
Kind regards,
All the Authors

Reviewer 2 Report
Comments and Suggestions for Authors
I think the Roughness (ra value) of the used and new burs should have been measured by laser surface scanner.
I think it would have been good to have a control group.
We should have been known that how much tissue is removed from the prepared tooth and it is also important to know how long the bur and tooth had in contact
Author Response
Dear Reviewer 2,
We thank you for revising our article and for you valuable suggestions. In what follows we have taken into consideration your guidelines as it can be seen in the attached answer.
Kind regards,
All the Authors

Reviewer 3 Report
Comments and Suggestions for Authors
The study seems interesting and genuine irrespective of high similarity index per the iThenticate report. The authors should address the following points to improve the quality of the manuscript:
- The abstract was well-written, concise and informative.
- Introduction section can be reduced to be concise and up to the point.
- Since the study is clinically oriented, have authors registered the study in clinical trial registry database?
- Inclusion and exclusion criteria were not discussed in the manuscript.
- Study protocol is rigorous and well-done. However, the authors should mention the type and caorseness of the diamond burs used in the study.
- Although 5 uses seems reasonable for standard protocol, did authors exmine the used burs as it may cause heterogeneity in the recorded data.
- How did authors control fluid control (suction) while tooth preparation and will this fluid measurement protocol?
- Discussion is short and authors should expand this section.
- Directions for future research should be added.
- Conclusion should be expanded and listed in bullet points.
Author Response
Dear Reviewer 3,
We would like to thank you for your appreciation of our work and say we are glad to hear that you found it interesting. We have taken into consideration your recommendations, therefore we have operated the following changes as it can be seen in the attached answer.
Kind regards,
All the Authors

Round 2
Reviewer 1 Report
Comments and Suggestions for Authors
Thank you for addressing all of the previous comments. I think that you clarified all the aspects.